# Systematic Review of Prehospital Prediction Models for Identifying Intracerebral Haemorrhage in Suspected Stroke Patients

**DOI:** 10.3390/healthcare13080876

**Published:** 2025-04-11

**Authors:** Mohammed Almubayyidh, Ibrahim Alghamdi, David Jenkins, Adrian Parry-Jones

**Affiliations:** 1Division of Cardiovascular Sciences, Faculty of Biology, Medicine and Health, University of Manchester, Manchester M13 9PL, UK; 2Department of Aviation and Marine, Prince Sultan Bin Abdulaziz College for Emergency Medical Services, King Saud University, Riyadh 145111, Saudi Arabia; 3Department of Emergency Medical Services, College of Applied Medical Sciences, Khamis Mushait Campus, King Khalid University, Abha 62521, Saudi Arabia; 4Division of Informatics, Imaging and Data Science, Faculty of Biology, Medicine and Health, University of Manchester, Manchester M13 9GB, UK; 5NIHR Manchester Biomedical Research Centre, Manchester University NHS Foundation Trust, Manchester Academic Health Science Centre, Manchester M13 9WL, UK; 6Geoffrey Jefferson Brain Research Centre, Manchester Academic Health Science Centre, Northern Care Alliance & University of Manchester, Manchester M13 9PL, UK

**Keywords:** prehospital, stroke, intracerebral haemorrhage, prediction, differential diagnosis

## Abstract

**Introduction:** The prompt prehospital identification of intracerebral haemorrhage (ICH) may allow very early delivery of treatments to limit bleeding. Current prehospital stroke assessment tools have limited accuracy for the detection of ICH as they were designed to recognise all strokes, not ICH specifically. This systematic review aims to evaluate the performance of prehospital models in distinguishing ICH from other causes of suspected stroke. **Methods:** We adhered to the Preferred Reporting Items for Systematic Reviews and Meta-Analyses guidelines. Following a predefined strategy, we searched three electronic databases via Ovid (MEDLINE, EMBASE, and CENTRAL) in July 2023 for studies published in English, without date restrictions. Subsequently, data extraction was performed, and methodological quality was assessed using the Prediction Model Risk of Bias Assessment Tool. **Results:** After eliminating duplicates, 6194 records were screened for titles and abstracts. After a full-text review of 137 studies, 9 prediction studies were included. Five of these described prediction models were designed to differentiate between stroke subtypes, three distinguished between ICH and ischaemic stroke, and one model was developed specifically to identify ICH. All studies were assessed as having a high risk of bias, particularly in the analysis domain. The performance of the models varied, with the area under the receiver operating characteristic curve ranging from 0.73 to 0.91. The models commonly included the following as predictors of ICH: impaired consciousness, headache, speech or language impairment, high systolic blood pressure, nausea or vomiting, and weakness or paralysis of limbs. **Conclusions:** Prediction models may support the prehospital diagnosis of ICH, but existing models have methodological limitations, making them unreliable for informing practice. Future studies should aim to address these identified limitations and include a broader range of suspected strokes to develop a practical model for identifying ICH. Combining prediction models with point-of-care tests might further improve the detection accuracy of ICH.

## 1. Introduction

Strokes are a leading cause of morbidity and mortality worldwide, posing a significant healthcare challenge. Among the different stroke subtypes, intracerebral haemorrhage (ICH) accounts for only 10–15% of all strokes, but it is the most devastating form with no significant improvement in outcomes [1,2]. Timely and accurate identification of ICH is needed to facilitate prompt and appropriate management, which may be associated with better patient outcomes [3].

Given that most strokes occur out of hospital, the prehospital phase plays a critical role in the stroke patient care pathway. While advances in screening tools have enabled the early detection of strokes in the prehospital stage, these efforts primarily focus on identifying ischaemic stroke (IS) or all strokes [4]. Previous studies have shown that these tools have limited accuracy in identifying patients with ICH [5,6]. This limitation arises from the tools originally being designed to predict any stroke, rather than specific stroke subtypes (ICH or IS).

In recent years, significant advancements have been made in diagnosing and treating patients with ICH in the prehospital setting through the use of mobile stroke units [3]. Nevertheless, due to the high associated costs, it is unlikely that this technology will become widely available for stroke care in the near future, highlighting the need for cost-effective alternatives to enhance prehospital stroke care.

It is clinically important to distinguish ICH from other suspected stroke cases in the prehospital setting, as certain time-sensitive interventions, such as lowering blood pressure and reversing coagulopathy, are critical for reducing the risk of haematoma expansion, a major cause of poor outcomes [3,4]. This is supported by recent findings from the INTERACT4 trial [7], which demonstrated that intensive blood pressure lowering, administered to suspected stroke patients in the ambulance, significantly reduced death and disability at 90 days among those subsequently diagnosed with ICH. However, the same intervention proved detrimental in patients with IS [7], underscoring the need for diagnostic certainty in the prehospital setting. Furthermore, patients with ICH may benefit from being transported to the nearest appropriate stroke centre to minimise treatment delays [8]. Longer transport times to a thrombectomy-capable centre—the preferred destination for patients with large vessel occlusion (LVO)—have been associated with poorer outcomes in patients later diagnosed with ICH [9]. Consequently, prehospital differentiation of ICH, particularly from IS and LVO, becomes a priority to determine the optimal management strategies for suspected stroke cases in the field. This systematic review therefore aims to evaluate and compare the performance of available prehospital stroke prediction models in distinguishing ICH from other causes of suspected stroke.

## 2. Methods

### 2.1. Protocol and Registration

The Preferred Reporting Items for Systematic Reviews and Meta-Analyses (PRISMA) guidelines were followed to conduct and report this systematic review [10]. A completed PRISMA checklist is provided in Appendix A. Following scoping searches, a protocol was developed for this review and registered with the International Prospective Register of Systematic Reviews (PROSPERO) on 9 August 2023 (registration number: CRD42023452526).

### 2.2. Eligibility Criteria

The inclusion and exclusion criteria are outlined in Table 1.

### 2.3. Search Strategy

Using a prespecified search strategy (Appendix A), a systematic literature search of the MEDLINE, EMBASE, and CENTRAL databases was conducted via Ovid from inception to July 2023. The search was limited to human studies, and for resource reasons, was restricted to studies published in English. After completing the database searches, a search of grey literature was conducted using Google Scholar. Additionally, the reference lists of included studies were reviewed to identify additional studies of interest.

### 2.4. Study Selection

The identified articles were imported into EndNote and Rayyan Qatar Computing Research Institute (QCRI) software [11], enabling duplicates to be eliminated, as well as facilitating screening and collaboration. Suitable studies were selected by two independent reviewers (MA and IA) in two stages. First, titles and abstracts were screened, and then full texts were carefully evaluated. MA screened all titles, abstracts, and full-text articles against the eligibility criteria. IA screened a random sample of 20% of the articles at both the title and abstract stage and the full-text stage. Disagreements during these screening stages were resolved through discussion between the reviewers to reach a consensus or by involving a third reviewer (DAJ or APJ) when necessary.

### 2.5. Data Extraction

The data were extracted from the eligible studies using standardised, prepiloted forms. MA extracted the data, and IA double-checked it for accuracy, with any differences being discussed and resolved. The following information was extracted, in accordance with the review objectives:First author, year, country;Study design;Study population demographics and baseline characteristics;Data collection setting;Details of the stroke prediction model and its variables;The reference standard used to determine the final diagnosis;Reported or calculated diagnostic accuracy metrics, including sensitivity, specificity, positive and negative predictive values, and area under the receiver operating characteristic curve (AUC) with their 95% confidence intervals (CI).

As this review only considered published data, the original study authors were not contacted for additional or missing information.

### 2.6. Risk of Bias and Applicability Assessment

Two reviewers used the Prediction Model Risk of Bias Assessment Tool (PROBAST) to evaluate the quality of the included studies [12]. PROBAST has been designed specifically to assess the risk of bias and applicability concerns in prediction modelling studies. To evaluate the risk of bias, 20 signalling questions across four domains were addressed, namely participants, predictors, outcome, and analysis. The applicability assessment consisted of several questions across three domains: participants, predictors, and outcome.

The overall assessment of the risk of bias and concerns regarding applicability was classified as ‘low’ if all domains received a ‘low’ rating, or ‘high’ if at least one domain was rated as ‘high’; ‘unclear’ was used where any domain was judged as such. The quality of the studies included in this review was summarised narratively and is presented in tabulated and graphical formats.

### 2.7. Data Synthesis

Due to the anticipated heterogeneity of populations, comparators, and outcome variables across the studies, a narrative synthesis approach was employed to synthesise the data. Additionally, the findings were summarised in tables and figures, and when appropriate, a comparison was made between the available models to identify similarities and differences, as well as assess their diagnostic accuracy performance. Unreported diagnostic metrics were calculated with the MedCalc statistical software, using the data provided in each study. For these metrics, 95% CIs were derived using the Clopper–Pearson method [13].

### 2.8. Patient and Public Involvement

No patients or public were involved in this study.

## 3. Results

### 3.1. Study Selection

A total of 8112 records were identified through the search in databases. After eliminating duplicate entries, 6194 unique records remained to be screened for titles and abstracts (Figure 1). Following the review of titles and abstracts, 137 articles were identified as being potentially relevant and were retrieved for a thorough evaluation of their full texts. After this evaluation, 128 articles were excluded for various reasons, including differences in patient populations, study types, settings, and reported outcomes, as well as one duplicate publication (illustrated in Figure 1). As a result, nine articles met our eligibility criteria [14,15,16,17,18,19,20,21,22].

### 3.2. Characteristics of the Included Studies

The characteristics of the included studies are outlined in Table 2. The majority of prediction models were developed in Asia (n = 6), followed by Europe (n = 2) and North America (n = 1). Additionally, these models were constructed based on either prospective (n = 4) or retrospective cohort data (n = 2), or a combination of both (n = 3). Most of the included models were designed to differentiate between stroke subtypes [17,18,19,21,22]; three of the models were developed to distinguish between ICH and IS [14,15,16]. Only one model focused specifically on distinguishing patients with ICH from other stroke subtypes and non-stroke diagnoses [20]. All of the included studies targeted patients in prehospital settings, and the majority used only prehospital information to develop their models (n = 6).

### 3.3. Risk of Bias and Applicability Assessment

Table 3 summarises the quality assessment of the included studies. The majority of models (n = 6) demonstrated a low risk of bias in the domains of participants, predictors, and outcome (Figure 2A). However, two models exhibited a high risk of bias in the participants domain due to their exclusion criteria, leading to only a selected group of patients with ICH and IS [14,15]. In the predictors domain, three models had an unclear risk of bias because of insufficient information about knowledge of outcomes before assessing predictors [14,15], or a lack of information about predictors [21]. Additionally, a high or unclear risk of bias was observed in two models in the outcome assessment domain, primarily due to the absence of a clear or standardised definition of ICH [14,15]. Furthermore, all studies had a high risk of analysis bias arising from various factors, including the limited sample size of participants with ICH in the model validation cohorts [16,20,21], selection of predictors based on univariable significance [15,16,17,19], inadequate exclusion of participants from the analysis [17,19,20,22], and the use of inappropriate performance measures [14,15,16,17,18,19]. By applying PROBAST, all models were considered at a high risk of overall bias, as shown in Figure 2A.

When the nine models were assessed for applicability concerns, seven of the models were determined to have an overall low concern after evaluating their applicability to the review questions (Table 3 and Figure 2B). Two models were rated as having unclear or high concerns on multiple domains and were judged to be of high applicability concern, either due to the definition of the outcome [14] or the intended outcome from the model [15].

### 3.4. Model Development and Final Predictor Variables

Among the nine studies included in the review, seven used multivariable analysis to derive their prediction rules, of which five presented scoring systems [14,15,17,19,20], one used conditional probability [16], and one combined the best performance predictors [18]. Additionally, two studies [21,22] used different machine learning algorithms, such as logistic regression, random forest, and eXtreme Gradient Boosting (XGBoost) to develop the prediction models.

To determine the probability of ICH and other diagnoses, the included studies combined between 2 [18] and 52 predictors [21] in their final models. These studies collected a wide range of prehospital and in-hospital data, as illustrated in Appendix A.

The predictors most frequently included in the final models were impaired consciousness (n = 8), headache (n = 6), speech or language deficit (n = 6), high systolic blood pressure (n = 6), nausea or vomiting (n = 5), and limb weakness or paralysis (n = 5). Other predictors are summarised in Appendix A.

Despite the overlap between predictors, specific associations between predictors and ICH were not reported in some studies [19,21,22]. Additionally, variations existed in how they were defined and assessed, making it difficult to directly compare their importance or weight across the various models. For instance, the level of consciousness and neurological deficits were assessed using the National Institutes of Health Stroke Scale [15,20], while different dichotomisations of blood pressure values were observed in the models [15,17,20].

### 3.5. Model Performance and Validation

Regarding performance metrics, the majority of the studies (n = 6) reported model discrimination using the AUC values [14,17,19,20,21,22], while other predictive characteristics were presented in four studies [18,20,21,22] and calculable in four of the nine studies [14,15,16,17] (as shown in Table 4). Furthermore, only two studies [19,22] demonstrated model calibration graphically with calibration plots, without further specification of the intercepts and slopes.

Three prediction models were developed to distinguish between ICH and IS [14,15,16]. Woisetschläger et al. [14] devised an out-of-hospital score, ranging from –3 to +3, where a positive score was predictive of ICH. The model achieved an AUC of 0.90 (95% CI, 0.86–0.94), with sensitivity ranging from 11% (95% CI, 6.0–18.1%) to 32% (95% CI, 23.9–41.4%) and specificity from 96% (95% CI, 90.6–99.0%) to 100% (95% CI, 96.6–100.0%) for cut-off scores between +1 and +3. Yamashita et al. [15] focused on distinguishing IS from ICH within 6 h of stroke onset. At a cut-off score of 0, their model had a sensitivity of 41% (95% CI, 31.3–51.3%) and a specificity of 91% (95% CI, 85.0–95.6%) for detecting ICH. Jin et al. [16] established a Bayes discriminant model for differentiating between ICH and IS in alert or comatose patients. In the training set, the predictive model showed moderate sensitivity (58%; 95% CI, 51.9–64.8%) and high specificity (79%; 95% CI, 74.8–83.2%) for diagnosing ICH in non-comatose patients, whereas in comatose patients with ICH, the sensitivity was high (94%; 95% CI, 90.2–96.0%), but specificity was low (42%; 95% CI, 34.5–49.8%).

Six prediction models were proposed to discriminate subtypes of stroke and other causes of stroke symptoms [17,18,19,20,21,22]. Uchida et al. [17] designed a prediction score to distinguish between different stroke subtypes simultaneously, including ICH, LVO, subarachnoid haemorrhage (SAH), and other types of strokes. This score consisted of 21 variables, which were then simplified into 7 variables by backward elimination [19]. In their derivation cohorts, the AUC for ICH ranged between 0.79 and 0.84 [17,19]. Similarly, Chiquete et al. [18] constructed different prediction rules to classify stroke subtypes based on clinical features recognised by bystanders who witnessed the onset of stroke. The best-performing model in this study achieved a sensitivity of 66% (95% CI, 57.0–74.6%) and a specificity of 52% (95% CI, 45.5–57.5%) in diagnosing patients with ICH [18].

Geisler et al. [20] developed the only prehospital prediction tool specifically for patients with ICH. According to the study, the likelihood of an ICH increased when the threshold score was ≥1.5. For the derivation cohort, the threshold scores exhibited low to moderate sensitivity ranging from 13% (95% CI, 3.5–29.0%) to 50% (95% CI, 31.9–68.1%), but high specificity ranging from 80% (95% CI, 75.9–84.1%) to 100% (95% CI, 98.6–100.0%), with an AUC of 0.75.

Uchida et al. [22] also employed three machine learning algorithms (logistic regression, random forest, and XGBoost) to develop a prehospital stroke scale. These algorithms demonstrated comparable predictive accuracy in distinguishing patients with ICH, with sensitivity ranging from 42% to 43%, specificity at 45% and AUC ranging from 0.78 to 0.79 in the training cohort. In contrast, the study conducted by Hayashi et al. [21] demonstrated superior predictive accuracy for ICH by using XGBoost to develop their prehospital model, achieving an AUC of 0.91 (95% CI, 0.89–0.93), with sensitivity and specificity values of 68% and 91%, respectively.

In terms of validation, six studies validated their prediction models, either by randomly dividing the study population into training and test sets [16,21] or by using an independent test cohort [17,19,20,22].

The model developed by Jin et al. [16] showed better accuracy in the validation cohort of non-comatose patients with ICH when compared to the derivation cohort, achieving a sensitivity of 66% (95% CI, 50.1–79.5%) and specificity of 87% (95% CI, 76.2–94.3%). However, its performance was comparatively lower in the comatose group, with a sensitivity of 91% (95% CI, 80.1–97.0%) and specificity of 39% (95% CI, 21.5–59.4%).

Geisler et al. [20] validated their model by comparing ICH and IS patients only. The sensitivity ranged from 3% (95% CI, 0.1–15.8%) to 52% (95% CI, 33.5–69.2%) and specificity from 87% (95% CI, 82.1–90.8%) to 100% (95% CI, 98.6–100.0%), with an AUC of 0.81.

The scoring systems devised by Uchida et al. were validated using the same outcome definitions as in their developed models [17,19]. The first model’s sensitivity and specificity for predicting ICH ranged from 2% (95% CI, 0.3–4.7%) to 33% (95% CI, 26.0–40.1%) and 46% (95% CI, 42.7–49.6%) to 98% (95% CI, 97.2–99.1%), respectively [17]. The AUC values for both models fell within the range of 0.73 and 0.77 [17,19].

The validation of the machine learning models employed by Uchida et al. [22] demonstrated better specificity (92–94%) with comparable sensitivity (40–43%), resulting in AUC values of 0.81 to 0.82. Similarly, Hayashi et al. [21] found that XGBoost had the highest performance among the classifiers, albeit with a slight decrease in sensitivity (62%), specificity (90%), and AUC value (0.87; 95% CI, 0.82–0.91) compared to the derivation set.

## 4. Discussion and Recommendations

This systematic review has identified nine prediction models for distinguishing ICH and other causes of suspected stroke in the prehospital setting. All the studies were deemed to have a high risk of bias. Furthermore, there was a considerable degree of heterogeneity in the study populations, designs, predictor variables, performance measures, and intended outcomes of the models. Consequently, we are unable to recommend the use of any of the studied models in the prehospital stroke care. Additionally, the external validity and generalisability of the models are unclear.

In the past, several attempts have been made to develop predictive scores for classifying stroke subtypes [23]. However, their diagnostic accuracy was poor, and they might not be applicable to patients in the prehospital setting [23,24]. Additionally, those scores were designed to distinguish between two main subtypes of stroke—ischaemic and haemorrhagic stroke—without considering other stroke-mimicking conditions; hence, they may not adequately capture the complexity of real-world care. Notably, only three of the included studies addressed the appropriate selection of patients for developing their prediction models, considering both stroke subtypes and other causes of suspected stroke [20,21,22].

The selection of an optimal prediction model is critical to support the recent advancements in ICH management [4,7]. Choosing models with high sensitivity will ensure the correct identification and triaging of patients with ICH to an appropriate level of care, yet it may lead to an increase in false positives. On the other hand, prioritising high specificity can prevent unnecessary treatment for non-ICH cases, but it might also result in a greater number of false negatives. Ideally, the ICH prediction model should balance high sensitivity and specificity. Nevertheless, it is important to recognise that the sensitivity and specificity trade off against one another. Notably, the included studies used different arbitrary cut-off points, which could influence the classification performances.

Among all the models studied, only one [21] achieved a good balance between sensitivity and specificity for detecting ICH using the XGBoost classifier, suggesting the superiority of machine learning models over traditional predictive models (Table 4). However, this model used 52 prediction features, limiting its practicality in the prehospital environment. In prehospital settings, it may be advisable to use prediction models with fewer variables, considering the diverse range of conditions that exist in practice other than stroke [22]. This approach was considered in most of the included studies, where a limited set of predictors was selected for their final models [14,15,16,18,19,20].

Most of the studies in this review collected common predictors of stroke. Among these, decreased consciousness level, high systolic blood pressure, headache, neurological deficits at presentation, and nausea or vomiting were more commonly included in the final models and should be considered as potential predictors for ICH in future prediction models. However, there were variations in the inclusion of other predictors (Appendix A), likely due to differences in the considered variables and study populations.

Through our search of the literature, we identified two other prehospital prediction models that were developed to identify the different subtypes of strokes [25,26]. However, these were not included in our review as they combined ICH and SAH into one category, which was defined as haemorrhagic stroke. ICH and SAH are separate disease entities with different pathophysiology, risk factors, clinical presentations, and management approaches [16,27]. Hence, it may be more beneficial for future models to consider the distinct characteristics of ICH and SAH, as their combination could potentially impact accurate diagnosis and appropriate treatment.

The findings of this systematic review highlight the need for further prehospital research to enhance the identification of ICH. Future studies can use our findings as a guide to develop predictive models that incorporate readily available prehospital data. To ensure their real-world applicability, it is important for future studies to consider a diverse cohort of suspected stroke patients, adhere to recent sample size criteria, and avoid data splitting for model development and validation [28]. Moreover, it would be more beneficial to validate and assess the accuracy, reliability, and limitations of the developed models by using large independent cohorts. Finally, future prediction studies should follow the TRIPOD (Transparent Reporting of a multivariable prediction model for Individual Prognosis Or Diagnosis) guidelines to ensure rigorous reporting [29].

Future prehospital studies might also consider using simple diagnostic aids in combination with predictive models to improve the accuracy of ICH detection [30,31]. These could include point-of-care testing technologies that measure brain-specific biomarkers, such as glial fibrillary acidic protein, which has shown promising results in distinguishing ICH from other conditions [32,33].

### Limitations

The current review has some limitations worth mentioning. First, our review was restricted to published studies in the English language, thus potentially introducing language and publication bias. Additionally, due to the high number of retrieved records, the second reviewer screened only 20% of the identified studies, which might have led to the exclusion of some potentially relevant research during the selection process. Lastly, the heterogeneity and methodological limitations of the included studies precluded the combination of data in a meta-analysis, therefore hindering our ability to provide recommendations for prehospital practice.

## 5. Conclusions

Existing prehospital stroke prediction models for distinguishing ICH and other causes of suspected stroke were heterogeneous and of poor methodological quality, precluding their recommendation for use in prehospital stroke care. With support from the findings of this review, future studies should address the identified limitations to develop and validate a practical prediction model for patients with ICH in the prehospital stage. In conjunction with predictive models, the use of point-of-care testing tools might further enhance the accuracy of ICH prediction and should be considered in future research.

## Figures and Tables

**Figure 1 healthcare-13-00876-f001:**
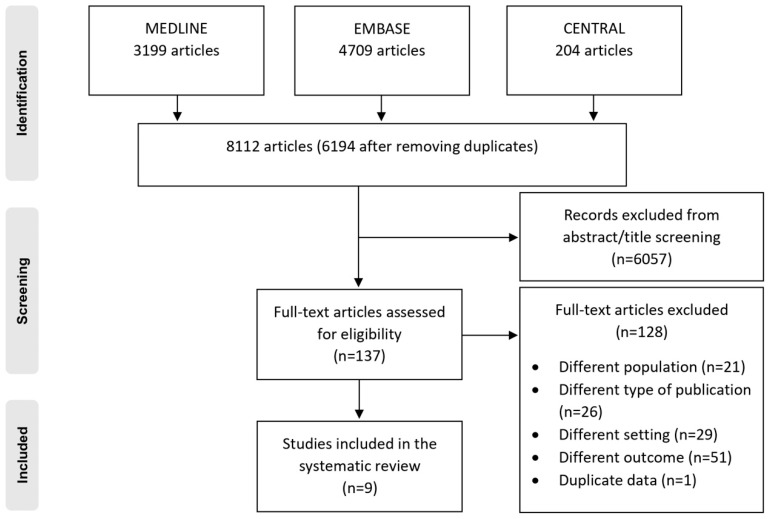
Flow chart of study selection process.

**Figure 2 healthcare-13-00876-f002:**
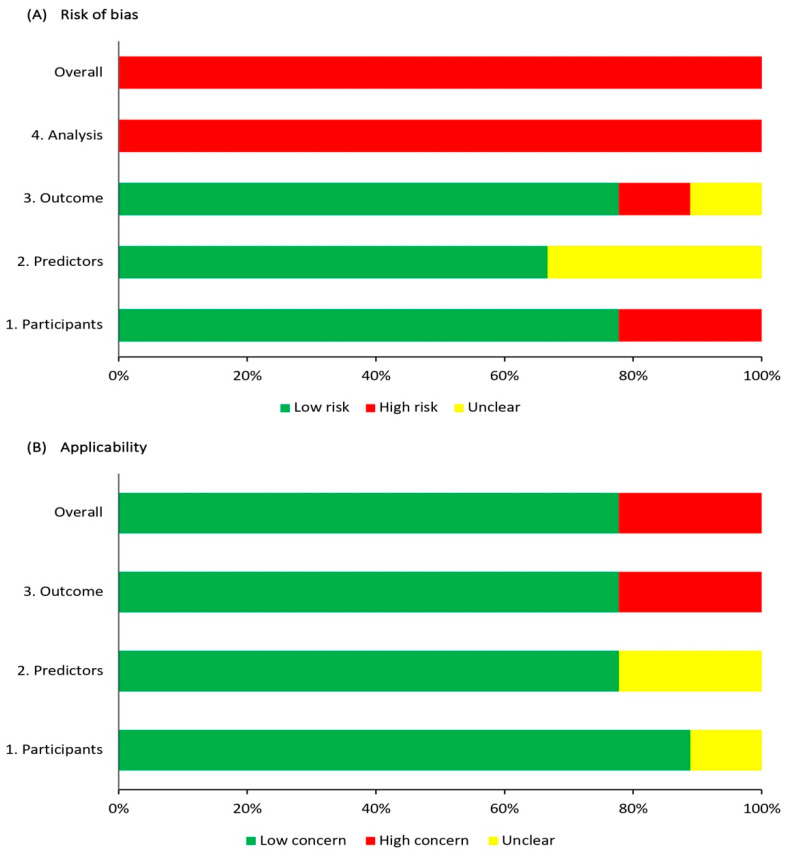
Summary of risk of bias (**A**) and applicability (**B**) assessment.

**Table 1 healthcare-13-00876-t001:** Inclusion and exclusion criteria.

**Inclusion Criteria**
Primary peer-reviewed research published in the English language, regardless of the year of publication. Adult patients (aged ≥ 16 years) with suspected stroke in the prehospital setting, confirmed by either CT or MRI scans. Studies that developed a prehospital prediction model to identify and distinguish patients with ICH from other causes of suspected stroke.
**Exclusion Criteria**
Reviews, conference abstracts, commentaries, surveys, case reports, non-human studies, and non-English studies. Studies involving children (aged < 16 years). Studies that combined spontaneous ICH cases with those of other aetiologies. Studies that integrated novel approaches, such as blood biomarkers and brain diagnostic devices, with clinical assessment in their prediction models. Studies that did not report the diagnostic accuracy of the developed models or provide calculable data.

Abbreviations: CT, computed tomography; ICH, intracerebral haemorrhage; MRI, magnetic resonance imaging.

**Table 2 healthcare-13-00876-t002:** Characteristics of included studies.

Reference, Year; Country	Study Design	Statistical Model Used	Validation Type	Sample Size (ICH Cases)	Name of the Prediction Model	Target Condition	Data Collection Setting	Gold Standard
Woisetschläger [14], 2000; Austria	Retrospective-prospective study	Multivariable logistic regression	NR	224 (118)	Out-of-hospital model	ICH and IS	Prehospital and in-hospital	CT scan
Yamashita [15], 2011; Japan	Retrospective study	Multivariable logistic regression	NR	227 (100)	KP3S	IS	Prehospital and in-hospital	CT or MRI scan
Jin [16], 2016; China	Prospective study	Multivariable logistic regression	Training and test split	DC: 1101 (547) VC: 189 (99)	NR	ICH and IS	Prehospital	CT or MRI scan
Uchida [17], 2018; Japan	Prospective study	Multivariable logistic regression	External cohort validation	DC: 1229 (169) VC: 1007 (183)	JUST score	Stroke subtypes	Prehospital	CT or MRI scan
Chiquete [18], 2021; Mexico	Prospective study	Multiple variable analysis	NR	369 (107)	NR	Stroke subtypes	Prehospital	CT or MRI scan
Uchida [19], 2021; Japan	Retrospective-prospective study	Multivariable logistic regression	External cohort validation	DC: 2236 (352) VC: 964 (138)	JUST-7 score	Stroke subtypes	Prehospital	CT or MRI scan
Geisler [20], 2021; Germany	Retrospective study	Multiple variable analysis	External cohort validation	DC: 416 (32) VC: 285 (33)	ph-ICH score	ICH	Prehospital and in-hospital	CT or MRI scan
Hayashi [21], 2021; Japan	Prospective study	ML algorithms: logistic regression, random forest, SVM, and XGBoost	Training and test split	DC: 1156 (271) VC: 290 (68)	NR	Stroke subtypes and non-stroke diagnoses	Prehospital	CT or MRI scan
Uchida [22], 2022; Japan	Retrospective-prospective study	ML algorithms: logistic regression, random forest, and XGBoost	External cohort validation	DC: 3178 (487) VC: 3127 (372)	JUST-ML	Stroke subtypes and non-stroke diagnoses	Prehospital	CT or MRI scan

Abbreviations: CT, computed tomography; DC, derivation cohort; ICH, intracerebral haemorrhage; IS, ischaemic stroke; JUST, Japan Urgent Stroke Triage; JUST-7, the 7-Item Japan Urgent Stroke Triage; JUST-ML, Japan Urgent Stroke Triage Score Using Machine Learning; KP3S, Kurashiki Prehospital Stroke Subtyping Score; ML, machine learning; MRI, magnetic resonance imaging; NR, not reported; ph-ICH, prehospital-intracerebral haemorrhage; SVM, support vector machine; VC, validation cohort; XGBoost, eXtreme Gradient Boosting.

**Table 3 healthcare-13-00876-t003:** Risk of bias and applicability assessment using the PROBAST tool.

Reference, Year	Risk of Bias				Applicability			Overall	
	Participants	Predictors	Outcome	Analysis	Participants	Predictors	Outcome	Risk of Bias	Applicability
Woisetschläger [14], 2000	−	?	−	−	?	?	−	−	−
Yamashita [15], 2011	−	?	?	−	+	?	−	−	−
Jin [16], 2016	+	+	+	−	+	+	+	−	+
Uchida [17], 2018	+	+	+	−	+	+	+	−	+
Chiquete [18], 2021	+	+	+	−	+	+	+	−	+
Uchida [19], 2021	+	+	+	−	+	+	+	−	+
Geisler [20], 2021	+	+	+	−	+	+	+	−	+
Hayashi [21], 2021	+	?	+	−	+	+	+	−	+
Uchida [22], 2022	+	+	+	−	+	+	+	−	+

(+): indicates low risk of bias/low concern regarding applicability. (−): indicates high risk of bias/high concern regarding applicability. (?): indicates unclear risk of bias/unclear concern regarding applicability. Abbreviations: PROBAST, Prediction Model Risk of Bias Assessment Tool.

**Table 4 healthcare-13-00876-t004:** Diagnostic performance of prehospital models for identifying patients with ICH.

Reference, Year	Comparison	Final Predictive Variables (Score)	Classification System	Sensitivity (95% CI)	Specificity (95% CI)	PPV (95% CI)	NPV (95% CI)	AUC (95% CI)
Woisetschläger [14], 2000	ICH vs. IS	Impaired LOC (+3). Hemisymptoms (−1). Other neurological symptoms (+1). History of DM (−1). History of HTN (−1).	−3	0%	93% (86.9–97.3)	0%	46% (44.4–46.9)	0.90 (0.86–0.94)
			−2	3% (0.9–8.5)	62% (52.3–71.5)	9% (3.6–21.3)	37% (33.2–40.3)	
			−1	10% (5.4–17.1)	70% (60.1–78.4)	27% (16.9–40.8)	41% (37.8–44.5)	
			0	18% (11.4–25.9)	81% (72.4–88.1)	51% (37.7–64.6)	47% (43.9–50.1)	
			+1	11% (6.0–18.1)	97% (92.0–99.4)	81% (55.9–93.7)	50% (47.7–51.3)	
			+2	25% (17.9–34.3)	96% (90.6–99.0)	88% (73.2–95.4)	54% (50.9–56.5)	
			+3	32% (23.9–41.4)	100% (96.6–100.0)	100% (90.8–100.0)	57% (53.9–60.0)	
Yamashita [15], 2011	ICH vs. IS	AF (+2). DBP < 100 mm Hg (+1). Lack of disturbance of consciousness (+1).	0	41% (31.3–51.3)	91% (85.0–95.6)	79% (66.9–87.3)	66% (62.3–70.0)	NR
			1	44% (34.1–54.3)	72% (63.8–80.0)	56% (46.8–64.3)	62% (57.3–66.8)	
			2	14% (7.9–22.4)	72% (63.0–79.3)	28% (18.2–40.5)	51% (48.0–54.8)	
			3/4	1% (0.0–5.5)	65% (55.6–72.9)	2% (0.3–13.7)	45% (42.1–48.6)	
Jin [16], 2016	ICH vs. IS	History of AF. Vomiting. History of DM. SBP ≥ 180 mm Hg. Age ≥ 65 years.	Non-comatose	DC: 58% (51.9–64.8) VC: 66% (50.1–79.5)	DC: 79% (74.8–83.2) VC: 87% (76.2–94.3)	DC: 63% (58.0–68.3) VC: 78% (64.7–87.8)	DC: 76% (72.6–78.5) VC: 78% (70.3–84.6)	NR
		Age ≥ 65 years. History of DM. History of AF. History of HTN. Vomiting.	Comatose	DC: 94% (90.2–96.0) VC: 91% (80.1–97.0)	DC: 42% (34.5–49.8) VC: 39% (21.5–59.4)	DC: 75% (72.3–77.2) VC: 75% (68.3–80.0)	DC: 78% (69.2–84.9) VC: 69% (45.9–85.1)	NR
Uchida [17], 2018	ICH vs. any stroke	History of IS (−2). Symptoms improved after onset (−2). Symptoms progressed after onset (+1). Headache (+1). Dysarthria (+2). SBP ≥ 165 mm Hg (+1). DBP ≥ 95 mm Hg (+1). Arrhythmia (−2). Disturbance of consciousness (+1). Conjugate deviation (+1). Paralysis of upper limbs (+2).	−6 to −2	VC: 2% (0.3–4.7)	VC: 88% (86.0–90.5)	VC: 3% (1.0–8.9)	VC: 80% (79.7–80.7)	DC: 0.84 VC: 0.77
			−1 to 2	VC: 21% (15.1–27.4)	VC: 46% (42.7–49.6)	VC: 8% (6.0–10.3)	VC: 72% (70.2–74.4)	
			3 to 4	VC: 33% (26.0–40.1)	VC: 77% (73.5–79.4)	VC: 24% (19.6–28.4)	VC: 84% (82.2–85.1)	
			5 to 6	VC: 31% (24.0–37.8)	VC: 91% (88.5–92.6)	VC: 42% (34.9–49.7)	VC: 85% (84.2–86.7)	
			7 to 9	VC: 14% (9.5–20.1)	VC: 98% (97.2–99.1)	VC: 65% (49.7–77.7)	VC: 84% (82.9–84.6)	
Chiquete [18], 2021	ICH vs. IS and SAH	Focal motor deficit + history of HTN.	N/A	66% (57.0–74.6)	52% (45.5–57.5)	36% (29.5–42.7)	79% (72.2–84.4)	NR
Uchida [19], 2021	ICH vs. any stroke	SBP ≥ 165 mm Hg. Arrhythmia. Conjugate deviation. Headache. Dysarthria. Disturbance of consciousness. Paralysis of upper limbs.	N/A	NR	NR	NR	NR	DC: 0.79 VC: 0.73
Geisler [20], 2021	DC: ICH vs. IS/TIA/SM VC: ICH vs. IS	SBP ≥ 180 mm Hg (+1). NIHSS LOC ≥ 1 (+1). NIHSS short (sum of the following NIHSS items: LOC, following commands, visual field, motor weakness of arm or leg, sensory disturbance divided by 10).	≥1.5	DC: 50% (31.9–68.1) VC: 52% (33.5–69.2)	DC: 80% (75.9–84.1) VC: 87% (82.1–90.8)	DC: 17% (12.4–23.9) VC: 34% (24.6–44.9)	DC: 95% (93.1–96.5) VC: 93% (90.6–95.1)	DC: 0.75 VC: 0.81
			≥2.0	DC: 38% (21.1–56.3) VC: 39% (22.9–57.9)	DC: 88% (84.4–91.1) VC: 94% (90.4–96.6)	DC: 21% (13.4–30.6) VC: 46% (31.2–62.4)	DC: 94% (92.8–95.7) VC: 92% (90.0–94.0)	
			≥2.5	DC: 28% (13.8–46.8) VC: 24% (11.1–42.3)	DC: 96% (93.6–97.8) VC: 98% (95.4–99.4)	DC: 38% (22.2–55.8) VC: 62% (35.7–82.2)	DC: 94% (92.8–95.2) VC: 91% (89.1–92.3)	
			≥3.0	DC: 25% (11.5–43.4) VC: 12% (3.4–28.2)	DC: 97% (95.3–98.7) VC: 100% (98.6–100.0)	DC: 44% (25.4–65.3) VC: 100% (39.8–100.0)	DC: 94% (92.7–95.0) VC: 90% (88.5–90.8)	
			≥3.5	DC: 13% (3.5–29.0) VC: 3% (0.1–15.8)	DC: 100% (98.6–100.0) VC: 100% (98.6–100.0)	DC: 80% (31.5–97.2) VC: 100% (2.5–100.0)	DC: 93% (92.3–94.0) VC: 89% (88.1–89.3)	
Hayashi [21], 2021	ICH vs. other strokes and non-stroke diagnoses	Age. Male sex. Past medical history: AF, HTN, DM, ICH, IS, epilepsy, psychiatric disorder, and anticoagulant use. Vital signs: HR, arrhythmia, SBP, DBP, temperature, JCS, and GCS. Symptoms: headache, vomiting, dizziness, convulsion, upper and lower limbs paralysis, hemiparalysis, conjugate deviation, visual field defects, facial palsy, ataxia, sensory impairment, aphasia, dysarthria, and unilateral spatial neglect. Onset timing Monday. Onset timing hours. Minimum THI.	XGBoost	DC: 68% VC: 62%	DC: 91% VC: 90%	NR	NR	DC: 0.91 (0.89–0.93) VC: 0.87 (0.82–0.91)
Uchida [22], 2022	ICH vs. other strokes and non-stroke diagnoses	Male sex. Age. SBP ≥ 165 mm Hg. DBP ≥ 95 mm Hg. Arrhythmia. Conjugate deviation. Aphasia. Headache. Convulsion. Dysarthria. Dizziness. Nausea or vomiting. Sudden onset. Symptoms improved after onset. Symptoms progressed after onset. Disturbance of consciousness. Facial palsy. Paralysis of upper limbs. Paralysis of lower limbs.	Logistic regression	DC: 43% VC: 43%	DC: 45% VC: 92%	DC: 46% VC: 42%	DC: 90% VC: 92%	DC: 0.79 VC: 0.82
			Random forest	DC: 42% VC: 41%	DC: 45% VC: 94%	DC: 50% VC: 46%	DC: 90% VC: 92%	DC: 0.79 VC: 0.82
			XGBoost	DC: 43% VC: 40%	DC: 45% VC: 92%	DC: 48% VC: 41%	DC: 90% VC: 92%	DC: 0.78 VC: 0.81

Abbreviations: AF, atrial fibrillation; AUC, area under the curve; CI, confidence interval; DBP, diastolic blood pressure; DC, derivation cohort; DM, diabetes mellitus; GCS, Glasgow Coma Scale; HR, heart rate; HTN, hypertension; ICH, intracerebral haemorrhage; IS, ischaemic stroke; JCS, Japan Coma Scale; LOC, level of consciousness; N/A, not applicable; NIHSS, National Institutes of Health Stroke Scale; NPV, negative predictive value; NR, not reported or cannot be calculated; PPV, positive predictive value; SAH, subarachnoid haemorrhage; SBP, systolic blood pressure; SM, stroke mimic; TIA, transient ischaemic attack; THI, thermo-hydrological index; VC, validation cohort; XGBoost, eXtreme Gradient Boosting.

## Data Availability

All data relevant to this review are included in the manuscript or uploaded as Appendix A.

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
