# Peer review of "Systematic Review of Prehospital Prediction Models for Identifying Intracerebral Haemorrhage in Suspected Stroke Patients"

_healthcare, 2025, doi:10.3390/healthcare13080876_

Round 1
Reviewer 1 Report
Comments and Suggestions for Authors
Thank you very much for the opportunity to review this paper.
The review was registered and reported according to PRISMA guidelines.
The authors have appropriately framed the research question and performed a systematic review of pre-hospital diagnostic models for ICH. Literature search was adequate (3 databases + grey literature), except for the restriction to studies in English. Authors recognize this limitation.
Study selection would have been more appropriate if performed 100% by two reviewers working independently. This limitation is also recognized by authors.
Authors adequately opted not to perform meta-analysis, due to the heterogeneity of included studies.
A formal risk of bias assessment using a validated tool is reported.
I only have three suggestions to improve the manuscript:
1- Since IA only screened a random sample of 20% of abstracts and papers for inclusion (lines 107-108), it would be nice to know the agreement above chance (kappa) for the two reviewers in those 20%. A high kappa coefficient would mean that it it is unlikely that studies were missed.
2- I suggest citing the INTERACT-4 trial (DOI: 10.1056/NEJMoa2314741). This study not only showed that early BP reduction is good for ICH patients, but also showed that it can be detrimental to ischemic stroke patients. Hence, adequate distinction between these two conditions in pre hospital care is fundamental.
3- Authors are pursuing prediction models for ICH diagnosis in the pre hospital care and refer to mobile stroke units as the "gold standard", never-the-less only available in a few locations in the globe. A separate line of investigation is focusing on AI and portable models using EEG, doppler or radiofrequencies. An interesting review is available in https://doi.org/10.3389/fneur.2024.1389056. AI using this biomarkers can constitute an important alternative to the method of interest in the systematic review for ICH diagnosis and author should discuss this possibility.
Author Response
Comment 1: Since IA only screened a random sample of 20% of abstracts and papers for inclusion (lines 107-108), it would be nice to know the agreement above chance (kappa) for the two reviewers in those 20%. A high kappa coefficient would mean that it it is unlikely that studies were missed.
Response 1: We thank the Reviewer for this insightful comment. We fully recognise the importance of measuring agreement in the screening process. However, due to the unavailability of screening records at this stage, we are unable to compute kappa retrospectively. Nonetheless, we would like to emphasise that our screening was conducted rigorously, following predefined inclusion and exclusion criteria to minimise bias and ensure consistency in the selection process. Furthermore, disagreements between the reviewers (MA and IA) were infrequent and were resolved through discussion, with a third reviewer involved on only one occasion. This high level of concordance suggests strong consistency and reliability in the screening process, making it unlikely that relevant studies were missed.
Comment 2: I suggest citing the INTERACT-4 trial (DOI: 10.1056/NEJMoa2314741). This study not only showed that early BP reduction is good for ICH patients, but also showed that it can be detrimental to ischemic stroke patients. Hence, adequate distinction between these two conditions in pre hospital care is fundamental.
Response 2: We thank the Reviewer for their valuable suggestion. The findings from the INTERACT4 trial have been cited and discussed in the Discussion and Recommendations section (lines 331–332). However, to further emphasise the importance of this study for the prehospital identification and management of ICH, we have added the following to our Introduction: “This is supported by recent findings from the INTERACT4 trial [7], which demonstrated that intensive blood pressure lowering, administered to suspected stroke patients in the ambulance, significantly reduced death and disability at 90 days among those subsequently diagnosed with ICH. However, the same intervention proved detrimental in patients with IS [7], underscoring the need for diagnostic certainty in the prehospital setting”.
Comment 3: Authors are pursuing prediction models for ICH diagnosis in the pre hospital care and refer to mobile stroke units as the "gold standard", never-the-less only available in a few locations in the globe. A separate line of investigation is focusing on AI and portable models using EEG, doppler or radiofrequencies. An interesting review is available in https://doi.org/10.3389/fneur.2024.1389056. AI using this biomarkers can constitute an important alternative to the method of interest in the systematic review for ICH diagnosis and author should discuss this possibility.
Response 3: We thank the Reviewer for raising this important point. While we agree that portable brain diagnostic devices hold promise for prehospital stroke care, our previous scoping review (DOI: 10.1136/bmjopen-2023-079316) examined their ability to detect ICH and found that most remain in the early stages of development, with few demonstrating the ability to distinguish patients with ICH. Moreover, none have yet been implemented in prehospital settings. Therefore, in the Discussion and Recommendations section (lines 372–376), we recommended that future studies explore the combination of predictive models with point-of-care devices that measure brain-specific biomarkers (e.g., GFAP). Such an approach may provide a low-cost, easy-to-use method for differentiating ICH from other conditions in prehospital settings and warrants investigation. We have now cited the suggested review alongside our scoping review for additional details regarding brain diagnostic devices (lines 372–373).
Reviewer 2 Report
Comments and Suggestions for Authors
This review aims to evaluate the performance of prehospital models in distinguishing ICH from other causes of suspected stroke. After review of 137 studies, nine prediction studies are examined in detail. It is found that all studies have a high risk of bias in the analysis domain and the performance of the models varied significantly. Overall, this manuscript is well written. The detail comments are as follows:
- The introduction is too simple. The authors should enrich the background about Prehospital Prediction for Identifying Intracerebral Haemorrhage. Are there any relevant practice that applies prediction models for Intracerebral Haemorrhage.
- This study only examined found 137 relevant articles. But 128 of them are excluded. The authors should explain reasons in detail for excluding those 128 papers in the text (not just in Figure 1).
- It would be better to incorporate some more general studies, which although not focus on prediction for Intracerebral Haemorrhage, are still useful and greatly cited and published in renowned top journals.
- The source journals of those relevant studies can be also presented.
- The authors can illustrate future research directions, inspired by the review study, with a framework that shows promising areas and incorporates the key features for developing more effective prediction models for Identifying Intracerebral Haemorrhage.
- The format of all tables can be improved. Using more concise three-line table can be better.
Author Response
Comment 1: The introduction is too simple. The authors should enrich the background about Prehospital Prediction for Identifying Intracerebral Haemorrhage. Are there any relevant practice that applies prediction models for Intracerebral Haemorrhage.
Response 1: Thank you very much for your valuable feedback. In line with your recommendation, we have further emphasised the importance of identifying patients with ICH in the prehospital setting, particularly in facilitating timely interventions (e.g., blood pressure lowering) and optimising transport decisions (e.g., direct transfer to the nearest appropriate stroke centre) (lines 68–84).
Comment 2: This study only examined found 137 relevant articles. But 128 of them are excluded. The authors should explain reasons in detail for excluding those 128 papers in the text (not just in Figure 1).
Response 2: Thank you very much for this insightful comment. The reasons for excluding these studies are now discussed in the Results section (lines 160–163), in addition to the explanation provided in Figure 1.
Comment 3: It would be better to incorporate some more general studies, which although not focus on prediction for Intracerebral Haemorrhage, are still useful and greatly cited and published in renowned top journals.
Response 3: Thank you for your suggestion. However, our review is specifically focused on prehospital prediction models designed to distinguish patients with ICH. While studies on prehospital stroke prediction—such as PRESTO (DOI: 10.1016/S1474-4422(20)30439-7) and ACT-FAST (DOI: 10.1161/STROKEAHA.117.019307)—may provide valuable insights into prehospital stroke care overall, they were not designed to differentiate ICH. Therefore, they do not meet our predefined eligibility criteria, which require a direct focus on ICH differentiation, given the current need to distinguish it from other suspected stroke cases. Including such studies would broaden the scope beyond our intended objective and may not provide insights directly relevant to the specific challenges of identifying ICH among suspected stroke cases in prehospital settings.
Comment 4: The source journals of those relevant studies can be also presented.
Response 4: Thank you for your suggestion. All included studies are fully cited in the tables, allowing readers to readily identify the publication sources from the reference list. Our review focuses on analysing the content and quality of the identified prediction models rather than the journals in which they were published. However, we have now clarified in Table 1 that only peer-reviewed primary research studies published in English were considered for inclusion.
Comment 5: The authors can illustrate future research directions, inspired by the review study, with a framework that shows promising areas and incorporates the key features for developing more effective prediction models for Identifying Intracerebral Haemorrhage.
Response 5: We thank the reviewer for this valuable suggestion. Although all the included studies had an overall high risk of bias, we have provided various recommendations in the Discussion and Recommendations section (lines 362–376) on how to address these methodological limitations in accordance with best practice guidelines (e.g., TRIPOD). Additionally, as mentioned in lines 372–376, future prediction models could also consider incorporating a simple diagnostic aid, such as point-of-care biomarker testing, to further enhance their accuracy. This could facilitate better decision-making for prehospital personnel when treating suspected stroke cases. We have now cited our previous review on these diagnostic technologies, which could be used in combination with prediction models (lines 372–373).
Comment 6: The format of all tables can be improved. Using more concise three-line table can be better.
Response 6: Thank you for your suggestion regarding table formatting. While a three-line table format is often effective for simpler tables, we tested this approach and found it difficult to include all the necessary information while maintaining readability. Our tables present detailed and structured data, including study characteristics, model performance metrics and risk of bias assessments, all of which require a clear format to ensure transparency and facilitate accurate comparisons. Adopting a three-line format made it more challenging to distinguish key information and increased the risk of misinterpretation. Therefore, we have retained the structured tables while making some modifications to improve spacing, alignment and overall readability in line with your suggestion.
Round 2
Reviewer 1 Report
Comments and Suggestions for Authors
Thank you very much for this opportunity, I am satisfied with the answers provided.